# Review of the Korean Species of the Genus *Edaphus* Motschulsky (Coleoptera, Staphylinidae) with Description of Four New Species [note 1]

**DOI:** 10.3390/insects13040362

**Published:** 2022-04-07

**Authors:** Ui-Joung Byeon, Sun-Jae Park, Seung-Gyu Lee, Jong-Seok Park

**Affiliations:** 1Chungbuk National University Insect Collection, Department of Biological Sciences and Biotechnology, Chungbuk National University, 1 Chungdae-ro, Seowon-gu, Cheongju-si 28644, Korea; dmlwjd1081@naver.com; 2Animal Resources Division, National Institute of Biological Resources, Incheon 22689, Korea; sun1763@korea.kr (S.-J.P.); jspdi84@korea.kr (S.-G.L.)

**Keywords:** beetle, new species, palaearctic, taxonomy, South Korea

## Abstract

**Simple Summary:**

Four new species, *Edaphus haenamensis*
**sp.n.**, *E. odaesanensis* **sp.n.**, *E. suyuensis* **sp.n**., and *E. ulsanensis* **sp.n.**, are described in the Korean Peninsula. Additional two species, *E. koreanus* and *E. lederi*, are redescribed, and the latter species is recorded for the first time from Korea.

**Abstract:**

The cosmopolitan euaesthetine genus, *Edaphus* Motschulsky, 1857, with about 90 Palaearctic species, was formerly known by a single species, *E. koreanus* Puthz, 2011, of which 2 specimens were collected in the southern part of the Korean Peninsula. In this paper, the knowledge of the Korean *Edaphus* fauna is expanded to include six species, including four described here based on a rich material collected in recent years. A key to all six species of Korean *Edaphus*, illustration of the habitus and diagnostic characters, and a distribution map are provided.

## 1. Introduction

The genus *Edaphus* Motschulsky, 1857 [1], is the largest genus of Euaesthetinae Thomson, 1859 [2], including about 600 species in the world [3,4]. More than half of the species are distributed in the Palaearctic and Oriental regions, and about 100 species are distributed in the Neotropical region, as well as the Afrotropical region [3]. Approximately 55 species have been recorded in Japan, but a single species, *E. koreanus* Puthz, 2011 [5], was recorded in the southern part of the Korean Peninsula [5,6]. During a revisionary study of the Korean *Edaphus* species based on 110 specimens, 4 new species and 1 additional, so far unrecorded species, were recognized. This is the first revisionary study of the Korean *Edaphus*, increasing the number of species identified in the Korean Peninsula to six.

## 2. Materials and Methods

All specimens were collected using sifting, mushroom window trap, flight intercept trap and lindgren funnel methods. At least a one specimen of each species was fully dissected to observe the male genitalia and other detailed features. These permanent microscopic slides were prepared using the method described by Hanley and Ashe [7]. The terminology and nomenclature were presented using the description by Chandler [8] and Puthz [3]. Numbering of abdominal sclerites indicated the morphological segment. The specimens were observed using a Leica M80 and MD 1000 LED optical microscope, and the images were generated using Las version 4.12 and Zerene stacker. The map of Korea was based on an image from SimpleMappr [9], which was subsequently modified to add locality marks. Specimen label data for holotypes were transcribed verbatim. Data for other specimens were standardized for consistency.

Holotypes of all species described herein are deposited in the National Institute of Biological Resources (NIBR, Incheon, Republic of Korea). Paratypes and voucher specimens are deposited in CBNUIC (Chungbuk National University Insect Collection, Cheongju, Republic of Korea) and NIBR (National Institute of Biological Resources, Incheon, Republic of Korea), indicated parenthetically. The following abbreviations were used (Figure 1):

## 3. Results


**Genus *Edaphus* Motschulsky, 1857**


*Edaphellus* Fauvel, 1878: 220 [10]

*Edaphus* LeConte, 1861: 67 [11]

*Hawkeswoodedaphus* Makhan, 2007: 1 [12]

*Microphthartus* Blattny, 1925: 185 [13]

*Rhenanus* Wusthoff, 1935: 48 [14]

*Tetratarsus* Schaufuss, 1877a: 24 [15]

*Tetrameres* Schaufuss, 1877b: 460 [16]

**Type species.***Edaphus nitidus* Motschulsky, 1857: 7 [1]

**Diagnosis.** Body with sparsely puncture, shining (Figure 2A, Figure 3A, Figure 4A, Figure 5A, Figure 6A and Figure 7A). Head with deep dorsal foveae between eyes (Figure 2B, Figure 3B, Figure 4B, Figure 5B, Figure 6B and Figure 7B); filiform antennae with 11 antennomere, 2- or 3-jointed club at the apex (Figure 2K, Figure 3K, Figure 4K, Figure 5K, Figure 6K and Figure 7K); mandibles small and thin (Figure 2N, Figure 3N, Figure 4N, Figure 5N, Figure 6N and Figure 7N); labrum with crenulated margin (Figure 2M, Figure 3M, Figure 4M, Figure 5M, Figure 6M and Figure 7M); maxillary palp with 4 palpomeres, IV extremely small (Figure 2L, Figure 3L, Figure 4L, Figure 5L, Figure 6L and Figure 7L). Tarsal formula 4-4-4 (Figure 2H–J, Figure 3H–J, Figure 4H–J, Figure 5H–J, Figure 6H–J and Figure 7H–J). First visible abdominal tergite with medio-basal carina and 1 pair of paratergites (Figure 2F, Figure 3F, Figure 4F, Figure 5F, Figure 6F and Figure 7F).

**Distribution.** Palaearctic, Oriental, Afrotropical, Australian, Nearctic, Neotropical, Pacific


***Edaphus haenamensis* Byeon, Park, Lee, and Park sp.n.**


(Figure 2, Figure 8A, Figure 9A and Figure 10)

**Type Material.** (n = 84, 58♂♂26♀♀), 58♂♂26♀♀ (16♂, genitalia dissected; 3♂♂1♀, slide mounted; 39♂♂25♀♀, dried). **Holotype.** 1♂, “**Korea**: Jeonnam Prov. Gurim-ri, Samsan-myeon, Haenam-gun, 19V2019, 34°29′42.7″ N 126°37′38.8″ E, 147 m, Sifting, Leaf litter and Deadwood debris J.-W. Kang″. **Paratypes.** Jeonnam: 1♂2♀♀ (1♂ slide mounted, CBNUIC), same data as holotype; 1♂2♀♀ (1♂ genitalia dissected and mounted in Euparal on plastic card, CBNUIC), Sinan-gun, Heuksan-myeon, Ye-ri, Mt. Chliac, 18 IV 2021, 34°40′39.3″ N 125°26′22.2″ E, 116 m, sifting pine tree debris, J.-W. Kang; 1♂ (1♂ genitalia dissected and mounted in Euparal on plastic card, CBNUIC), Wando-gun, Bogil-myeon, Buhwang-ri, 25 V 2021, 34°14′82.0″ N 126°54′72.0″ E, 76 m, sifting leaf litter and soil, J.-W. Seo; 1♂ (1♂ genitalia dissected and mounted in Euparal on plastic card, CBNUIC), Yeongam-gun, Haksan-myeon, Hakgye-ri, 19 V 2019, 34°40′44.9″ N 126°37′10.9″ E, 160 m, sifting mushroom and leaf litter and plant root under rock in bamboo forest, S.-H. Choi, U.-J. Byeon; Jeonbuk: 1♂1♀ (1♂ genitalia dissected and mounted in Euparal on plastic card, CBNUIC), Gunsan-si, Seongsan-myeon, Yeobang-ri, 03 VIII 2021, 36°00′38.1″ N 126°47′00.9″ E, 175 m, sifting soil and leaf litter, J.-W. Kang, J.-W. Kim; 5♂♂2♀♀ (1♂ genitalia dissected and mounted in Euparal on plastic card, CBNUIC), Iksan-si, Samgi-myeon, Seongnam-ri, 03 VIII 2021, 36°02′07.4″ N 127°01′54.3″ E, 244 m, sifting soil and leaf litter, U.-J. Byeon, M.-H. Song, J.-W. Seo; 12♂♂7♀♀, Yeondong-ri, 03 VIII 2021, 36°01′58.1″ N 127°02′11.6″ E, 346 m, sifting soil and leaf litter, U.-J. Byeon, M.-H. Song, J.-W. Seo; Chungbuk: 1♀, Danyang-gun, Danyang-eup, Dangdong-ri 23-1, 22 V 2020, 36°55′38.9″ N 128°22′30.9″ E, 348 m, sifting leaf litter and soil, T.-Y. Jang, Y.-J. Choi; 1♂1♀ (1♂ slide mounted, CBNUIC), Danyangsimgok-ri, 22 V 2020, 36°57′22.0″ N 128°20′40.4″ E, 225 m, sifting leaf litter and soil, U.-J. Byeon; 1♂1♀ (1♀ slide mounted, CBNUIC), Simgok-ri 330, 22 V 2020, sifting leaf litter and soil, U.-J. Byeon; 1♂, Yangbangsan-gil, 21 V 2020, 36°58′14.2″ N 128°22′57.6″ E, 585 m, sifting leaf litter and soil, Y.-J. Choi, U.-J. Byeon; 1♂ (1♂ genitalia dissected and mounted in Euparal on plastic card, CBNUIC), Goesan-gun, Yeonpung-myeon, Galgeum-ri, 30 V 2021, 36°78′57.0″ N 127°96′32.6″ E, 295 m, sifting leaf litter and soil, J.-W. Seo; 2♂♂ (1♂ genitalia dissected and mounted in Euparal on plastic card, CBNUIC), Boeun-gun, Songnisan-myeon, Biryongdongwan-ro, 22 VII 2021, 36°29′37.0″ N 127°51′12.0″ E, 280 m, sifting leaf litter & soil, Y.-J. Choi, J.-W. Kim; 1♂, Cheongju-si, Seowon-gu, Chungdae-ro, 17 VI 2021, 36°37′43.9″ N 127°27′18.8″ E, 70 m, sifting leaf litter and soil, M.-H. Song, J.-I. Shin; 1♀, Cheongwon-gu, Bui-myeon, Hwasang-ri, 16 VI 2020, 36°44′08.0″ N 127°29′01.4″ E, 38 m, sifting dead herbal stem and soil, T.-Y. Jang; 1♂1♀ (1♂ genitalia dissected and mounted in Euparal on plastic card, CBNUIC), Jecheon-si, Hansu-myeon, Songgye-ri, 09 VI 2021, 36°52′53.3″ N 128°05′13.8″ E, 292 m, sifting leaf litter and soil and root, M.-H. Song, J.-I. Shin; Chungnam: 2♂♂ (1♂ slide mounted, CBNUIC), Boryeong-si, Seongju-myeon, Seongju-ri, 26 V 2018, 327 m, sifting leaf litter, Y.-J. Choi; Daejeon Metropolitan City: 1♂ (1♂ genitalia dissected and mounted in Euparal on plastic card, CNUIC), Yuseong-gu, Eoeun-dong, Chungnam National University, 27 V 2002, sifting, J.-S. Park, J.-H. Choi; Gyeonggi: 2♂♂ (1♂ genitalia dissected and mounted in Euparal on plastic card, CBNUIC), Paju-si, Gwagtan-myeon, Yeongjang-ri, 15 V 2021, 37°45′06.0″ N 126°54′56.0″ E, 190 m, sifting leaf litter and soil, J.-W. Seo; 1♀, Yangju-si, Jangheung-myeon, Uldae-ri, 09 VII 2019, 37°42′45.0″ N 126°59′10.0″ E, 170 m, sifting leaf and deadwood debris near stream, Y.-J. Choi, T.-Y. Jang; Gangwon: 1♂ (1♂ genitalia dissected and mounted in Euparal on plastic card, CBNUIC), Gangneung-si, Gangdong-myeon, Dangyeong-ro, 13 VII 2021, 37°40′14.4″ N 128°55′05.7″ E, 172 m, sifting leaf litter and soil near stream, M.-H. Song; 1♂ (1♂ genitalia dissected and mounted in Euparal on plastic card, CBNUIC), Pyeongchang-gun, Jinbu-myeon, Odaesan-ro, 08 V 2021, 37°43′42.2″ N 128°35′48.0″ E, 666 m, sifting soil and leaf litter and deadwood, J.-W. Seo; Gyeongbuk: 2♂♂ (1♂ genitalia dissected and mounted in Euparal on plastic card, CBNUIC), Cheongdo-gun, Unmun-myeon, Seoji-ri, 02 VII 2021, 35°45′18.9″ N 128°56′41.5″ E, 251 m, sifting leaf litter and soil, M.-H. Song, U.-J. Byeon, J.-I. Shin; 2♂♂, Gyeongsan-si, Yongseong-myeon, Buil-ri, 02 VII 2021, 35°47′35.3″ N 128°55′30.4″ E, 406 m, sifting leaf litter and soil, M.-H. Song, U.-J. Byeon, J.-I. Shin; Gyeongnam: 5♂♂ (1♂ genitalia dissected and mounted in Euparal on plastic card, CBNUIC), Changyeong-gun, Yueo-myeon, Daedae-ri, 06 X 2021, 35°33′13.0″ N 128°25′24.0″ E, 20 m, sifting soil and herb debris, Y.-J. Choi, J.-I. Shin; Jeju: 1♂, Seoguipo-si, 1100-ro, 791, Georinsaseum platform, 20 VII 2021, J.-S. Oh, J.-W. Son, W.-W. Kim, S.-W. Yun; 1♂3♀♀, Sanghyo-dong, Donnaeko-ro, 26 IX 2021, 33°17′59.8″ N 126°34′59.3″ E, 269 m, sifting soil and leaf litter near waterfall, J.-W. Kang, U.-J. Byeon, T.-Y. Jang; 2♂♂3♀♀ (1♂ genitalia dissected and mounted in Euparal on plastic card, CBNUIC), sifting soil and leaf litter, J.-W. Kang, U.-J. Byeon; 1♂, Namwon-eup, Hannam-ri, 27 IX 2021, 33°20′56.9″ N 126°40′39.4″ E, 402 m, sifting deadwood debris, J.-W. Kang, U.-J. Byeon, T.-Y. Jang; 3♂, Sumang-ri, 27 IX 2021, 33°20′47.1″ N 126°40′36.4″ E, 322 m, sifting soil and leaf litter, J.-W. Kang, U.-J. Byeon; 2♂♂, 516-ro, 26 VIII 2021, 33°19′57.9″ N 126°36′25.2″ E, 504 m, sifting soil and deadwood, J.-W. Kang, J.-W. Kim, J.-I. Shin; Ulsan Metropolitan City: 1♂ (1♂ genitalia dissected and mounted in Euparal on plastic card, CBNUIC), Ulju-gun, Sangbuk-myeon, Deungeog-ri, 28 VI 2021, 35°33′10.5″ N 129°03′56.2″ E, 353 m, sifting leaf litter and soil, J.-W. Kang.

**Diagnosis.** This species can be distinguished from other *Edaphus* species by the following combination of characters: body reddish brown (Figure 2A); temples of head prominent (Figure 2B, arrow); antennomere X wider than length (width:length = 6.7:3.9, unit: 0.0085 mm), XI as long as it is wide (width:length = 6.7:7.1, unit: 0.0085 mm) (Figure 2K); pronotum with six medio-basal foveae (Figure 2C), medio-basal carina about twice longer than the medio-basal carina of tergite III (mbc:mbct3 = 9.2:4.8, unit: 0.0085 mm); elytron with three basal elytral foveae (Figure 2D); median lobe of male aedeagus 2.3 times longer than wide, paramere long seta twice longer than short seta (Figure 9A).

**Description.** Proportional measurements of habitus: HW: 29.9; DE: 18.4; LE: 8.1; LG: 24.8; LT: 28.1; PL: 30.6; PW: 34.7; dlbc: 22.5; EL: 36.5; EW: 43.8; mbc: 9.2; mbct3: 4.8 (unit: 0.0085 mm). Body length 1.2–1.7 mm (forebody length: 0.75 mm).

Head rectangular, vertex expanded, eyes convex (Figure 2B). Antennae bearing from edge of frons, I elongate, II–IV subrectangular, V–VI subquadrate, VII–IX trapezoid, X–XI club form (Figure 2K). Maxillary palpomeres I–II elongate, III longest (Figure 2L). Labrum with 7 teeth and 10 to 20 setae (Figure 2M). Mandible falciform and with two setae and eight teeth at mid-level (Figure 2N).

Pronotum wider than head (head:pronotum = 29.9:34.7, unit: 0.0085 mm) and wider anteriorly and with lateral basal carina and medio-basal carina (Figure 2C). Mesosternum with lateral mesocoxal foveae, basisternum with longitudinal carina and transverse carina (Figure 2E). Elytra wider than pronotum (elytra:pronotum = 43.8:34.7, unit: 0.0085 mm), elytron with basal elytral sulcus, distinct subhumeral elytral fovea and sulcus (Figure 2D).

Abdominal sternite III with basolateral foveae and short median longitudinal carina, III longest, IV–VII similar length (Figure 2G). Male abdominal sternite IX exposed on VIII, female not, VIII deeply emarginated posteriorly. Abdominal tergites III–VII with basolateral foveae (Figure 2F).

Aedeagus as in Figure 9A and internal sacs of median lobe movable.

**Distribution.** South Korea (Figure 10: circle).

**Etymology.** This species is named after the type locality, Haenam-gun, Jeonnam Province.

**Habitat.** Specimens of *E. haenamensis*
**sp.n.** were collected by sifting leaf litter and soil in the forest.


***Edaphus odaesanensis* Byeon, Park, Lee, and Park sp.n.**


(Figure 3, Figure 8B, Figure 9B and Figure 10)

**Type Material.** (n = 4, 4♂♂), 4♂♂ (3♂♂, genitalia dissected; 1♂, slide mounted). **Holotype.** 1♂ (1♂ genitalia dissected and mounted in Euparal on plastic card), “KOREA: Gangwon prov. Pyeongchang, Jinbu, Mt. Odae, Sangwonsa, 21 IV ~ 18 V 2002, S J Park, C W Shin, *ex* FIT”. **Paratypes.** Gangwon: 2♂♂ (1♂ genitalia dissected and mounted in Euparal on plastic card, 1♂ slide mounted, CNUIC), Pyeongchang-gun, Jinbu- myeon, Mt. Odaesan, sangwonsa temple, 18 VI 2004, S.-J. Park, *ex* FIT; Jeonbuk: 1♂ (1♂ genitalia dissected and mounted in Euparal on plastic card), Sinan-gun, Heksan-myeon, Gageo Island, Mt. Doksilsan, 13 VIII 2021, 34°04′51.9″ N 125°06′22.7″ E, 514 m, sifting soil & leaf litter, J.-W. Seo.

**Diagnosis.** This species can be distinguished from other *Edaphus* species by the following combination of characters: body reddish brown, abdomen dark brown (Figure 3A); temples of head oblique (Figure 3B); antennomere X as long as wide (width:length = 5.6:5.3, unit: 0.0085 mm), XI longer than wide (width:length = 5.6:7.9, unit: 0.0085 mm) (Figure 3K); pronotum with six medio-basal foveae (Figure 3C), medio-basal carina approximately 2.6 times longer than medio-basal carina of tergite III (mbc:mbct3 = 9.4:3.5, unit: 0.0085 mm); elytron with one basal elytral fovea (Figure 3D); median lobe of male aedeagus 2.5 times longer than wide, paramere long seta twice longer than short seta (Figure 9B).

**Description.** Proportional measurements of habitus: HW: 30.0; DE: 18.7; LE: 8.6; LG: 22.6; LT: 26.2; PL: 29.1; PW: 33.4; dlbc: 20.6; EL: 42.4; EW: 46.4; mbc: 9.4; mbct3: 3.5 (unit: 0.0085 mm). Body length 1.0–1.3 mm (forebody length: 0.72 mm).

Head rectangular, vertex expanded, eyes convex (Figure 3B). Antennae bearing from edge of frons, I–II elongate, III–VI subrectangular, VII subquadrate, VIII–IX trapezoid, X–XI club form (Figure 3K). Maxillary palpomeres I–II elongate, III longest (Figure 3L). Labrum with seven teeth and ten setae (Figure 3M). Mandible falciform and with two setae and seven teeth at mid-level (Figure 3N).

Pronotum wider than head (head:pronotum = 30.0:33.4, unit: 0.0085 mm) and wider anteriorly and with lateral basal carina and medio-basal carina (Figure 3C). Mesosternum with lateral mesocoxal foveae, basisternum with longitudinal carina and transverse carina (Figure 3E). Elytra wider than pronotum (elytra:pronotum = 46.4:33.4, unit: 0.0085 mm), elytron with basal elytral sulcus, distinct subhumeral elytral fovea and sulcus (Figure 3D).

Abdominal sternite III with basolateral foveae and short median longitudinal carina, III longest, IV–VII similar length (Figure 3G). Male abdominal sternite IX exposed on VIII, female unknown, VIII deeply emarginated posteriorly. Abdominal tergites III–VII with basolateral foveae (Figure 3F).

Aedeagus as in Figure 9B and internal sacs of median lobe movable.

**Distribution.** South Korea (Figure 10: triangle).

**Etymology.** This species is named after the type locality, Mt. Odae, Gangwon Province.

**Habitat.** Specimens of *E. odaesanensis*
**sp.n.** were collected by flight intercept trap (FIT) and sifting of leaf litter and soil in the forest.


***Edaphus suyuensis* Byeon, Park, Lee, and Park sp.n.**


(Figure 4, Figure 8C, Figure 9C and Figure 10)

**Type Material.** (n = 7, 4♂♂3♀♀), 4♂♂3♀♀ (1♂, genitalia dissected; 2♂♂1♀, slide mounted; 1♂2♀♀, dried). **Holotype.** 1♂ (1♂ genitalia dissected and mounted in Euparal on plastic card), “Korea: Seoul Suyu-dong, Gangbuk-gu, 07VII2019, 37°38′13.0″ N 126°59′41.0″ E, 280 m, Sifting, Leaf litter, T.-Y. Jang”. **Paratypes.** Seoul: 1♂1♀ (1♂ slide mounted, CBNUIC), same data as holotype; Chungbuk: 1♀, Cheongju-si, Seowon-gu, Gaesin-dong, 27 IV–21 V 2021, 36°37′42.2″ N 127°27′14.1″ E, 69 m, MWT (Mushroom Window Trap), T.-Y. Jang; 1♂1♀ (1♂1♀ slide mounted, CBNUIC), Chungdae-ro, 20 IV 2020, 36°37′43.5″ N 127°27′16.7″ E, 75 m, sifting soil and leaf litter and deadwood, T.-Y. Jang; 1♂, 14 V 2020, 36°37′46.5″ N 127°27′39.1″ E, 72 m, sifting soil and leaf litter, T.-Y. Jang, U.-J. Byeon, Y.-D. Choi.

**Diagnosis.** This species can be distinguished from other *Edaphus* species by the following combination of characters: body reddish brown (Figure 4A); temples of head prominent (Figure 4B, arrow); antennomere X wider than long (width:length = 5.2:4.2, unit: 0.0085 mm), XI longer than wide (width:length = 5.5:6.8, unit: 0.0085 mm) (Figure 4K); pronotum with six basal foveae (Figure 4C), medio-basal carina 1.3 times longer than medio-basal carina of tergite III (mbc:mbct3 = 8.2:5.9, unit: 0.0085 mm); elytron with three elytral foveae (Figure 4D); median lobe of male aedeagus twice longer than wide; paramere long seta about 1.3 times longer than short seta (Figure 9C).

**Description.** Proportional measurements of habitus: HW: 25.8; DE: 16.2; LE: 8.1; LG: 19.2; LT: 23.3; PL: 26.2; PW: 28.6; dlbc: 17.2; EL: 39.3; EW: 47.1; mbc: 8.2; mbct3: 5.9 (unit: 0.0085 mm). Body length 1.0–1.3 mm (forebody length: 0.67 mm).

Head rectangular, vertex expanded, eyes convex (Figure 4A). Antennae bearing from edge of frons, I–II elongate, III–VI subrectangular, VII subquadrate, VIII–IX trapezoid, X–XI club form (Figure 4K). Maxillary palpomeres I–II elongate, III longest (Figure 4L). Labrum with 7 teeth and 10 setae (Figure 4M). Mandible falciform and with two setae and nine teeth at mid-level (Figure 4N).

Pronotum wider than head (head:pronotum = 25.8:28.6, unit: 0.0085 mm) and wider anteriorly and with lateral basal carina and medio-basal carina (Figure 4C). Mesosternum with lateral mesocoxal foveae, basisternum with longitudinal carina and transverse carina (Figure 4E). Elytra wider than pronotum (elytra:pronotum = 47.1:28.6, unit: 0.0085 mm), elytron with basal elytral sulcus, distinct subhumeral elytral fovea and sulcus (Figure 4D).

Abdominal sternite III with basolateral foveae and median longitudinal carina, III longest, IV–VII similar length (Figure 4G). Male abdominal sternite IX exposed on VIII, female not, VIII deeply emarginated posteriorly. Abdominal tergites III–VII with basolateral foveae (Figure 4F).

Aedeagus as in Figure 9C and internal sacs of median lobe movable.

**Distribution.** South Korea (Figure 10: square).

**Etymology.** This species is named after the type locality, Suyu-dong, Seoul.

**Habitat.** Specimens of *E. suyuensis* were collected by sifting of leaf litter and soil in the forest, mushroom window trap (MWT) also used.


***Edaphus ulsanensis* Byeon, Park, Lee, and Park sp.n.**


(Figure 5, Figure 8D, Figure 9D and Figure 10)

**Type Material.** (n = 1, 1♂), 1♂ (1♂, slide mounted). **Holotype.** 1♂ (1♂ slide mounted), “Korea: Ulsan Metropolitan City, Deungeog-ri, Sangbuk-myeon, Ulju-gun, 28VI2021, 35°33′09.5″ N 129°03′56.3″ E, 386 m, Sifting, Soil and Leaf litter, J.-W. Kang”.

**Diagnosis.** This species can be distinguished from other *Edaphus* species by the following combination of characters: body bright reddish brown (Figure 5A); temples of head oblique (Figure 5B); antennomere X wider than long (width:length = 5.6:5.1, unit: 0.0085 mm), XI longer than wide (width:length = 5.8:7.8, unit: 0.0085 mm) (Figure 5K); pronotum with six basal foveae (Figure 5C); medio-basal carina about 1.6 times longer than medio-basal carina of tergite III (mbc:mbct3 = 9.4:5.6, unit: 0.0085 mm); elytron with one basal elytral fovea (Figure 5D); median lobe of male aedeagus 2.3 times longer than wide, paramere long seta 1.3 times longer than short seta (Figure 9D).

**Description.** Proportional measurements of habitus: HW: 29.4; DE: 20.0; LE: 8.2; LG: 21.2; LT: 25.9; PL: 28.2; PW: 34.1; dlbc: 20.2; EL: 43.5; EW: 44.2; mbc: 9.4; mbct3: 5.6 (unit: 0.0085 mm). Body length 1.1–1.4 mm (forebody length: 0.74 mm).

Head rectangular, vertex expanded, eyes convex (Figure 5B). Antennae bearing from edge of frons, I–II elongate, III–VI subrectangular, VII subquadrate, VIII–IX trapezoid, X–XI club form (Figure 5K). Maxillary palpomeres I–II elongate, III longest (Figure 5L). Labrum with 7 teeth and 10 setae (Figure 5M). Mandible falciform and with two setae and eight teeth at mid-level (Figure 5N).

Pronotum wider than head (head:pronotum = 29.4:34.1, unit: 0.0085 mm) and wider anteriorly and with lateral basal carina and medio-basal carina (Figure 5C). Mesosternum with lateral mesocoxal foveae, basisternum with longitudinal carina and transverse carina (Figure 5E). Elytra wider than pronotum (elytra:pronotum = 44.2:34.1, unit: 0.0085 mm), elytron with basal elytral sulcus, distinct subhumeral elytral fovea and sulcus (Figure 5D).

Abdominal sternite III with basolateral foveae and median longitudinal carina, III longest, IV–VII similar length (Figure 5G). Male abdominal sternite IX exposed on VIII, female unknown, VIII deeply emarginated posteriorly. Abdominal tergites III–VII with basolateral foveae (Figure 5F).

Aedeagus as in Figure 9D and internal sacs of median lobe movable.

**Distribution.** South Korea (Figure 10: reverse triangle).

**Etymology.** This species is named after the type locality, Ulsan Metropolitan City.

**Habitat.** Specimen of *E. ulsanensis*
**sp.n.** was collected by sifting of leaf litter and soil.


***Edaphus lederi* Eppelsheim, 1878 [17]**


(Figure 6, Figure 8E, Figure 9E and Figure 10)

**Material examined.** (n = 8, 6♂♂2♀♀), 6♂♂2♀♀ (1♂, slide mounted; 5♂♂2♀♀, dried). Jeju island: 6♂♂2♀♀ (1♂ slide mounted, CBNUIC), Jeju-si, Jocheon-eup, Bijarim-ro, 27 IX 2021, 33°25′47.6″ N 126°42′03.1″ E, 389 m, sifting reed near horse ranches, J.-W. Kang, U.-J. Byeon, T.-Y. Jang.

**Diagnosis.** This species can be distinguished from other *Edaphus* species by the following combination of characters: body dark brown (Figure 6A); temples of head prominent (Figure 6B, arrow); antennomere X wider than long (width:length = 4.8:3.1, unit: 0.0085 mm), XI longer than wide (width:length = 4.4:5.4, unit: 0.0085 mm) (Figure 6K); pronotum with six basal foveae (Figure 6C); medio-basal carina about 1.2 times longer than medio-basal carina of tergite III (mbc:mbct3 = 7.1:5.8, unit: 0.0085 mm); elytron with three basal elytral foveae (Figure 6D); median lobe of male 2.3 times longer than wide; paramere with two setae, long seta two times longer than short seta (Figure 9E).

**Redescription.** Proportional measurements of habitus: HW: 24.7; DE: 15.3; LE: 6.8; LG: 18.7; LT: 22.8; PL: 23.5; PW: 28.2; dlbc: 17.6; EL: 36.5; EW: 38.8; mbc: 7.1; mbct3: 5.8 (unit: 0.0085 mm). Body length 1.0–1.2 mm. (forebody length: 0.64 mm).

Head rectangular, vertex expanded, eyes convex (Figure 6B). Antennae bearing from edge of frons, I elongate, II–V subrectangular, VI subquadrate, VII–IX trapezoid, X–XI club form (Figure 6K). Maxillary palpomeres I–II elongate, III longest (Figure 6L). Labrum with 7 teeth and 20 setae (Figure 6M). Mandible falciform and with two setae and seven teeth at mid-level (Figure 6N).

Pronotum wider than head (head:pronotum = 24.7:28.2, unit: 0.0085 mm) and wider anteriorly and with lateral basal carina and medio-basal carina (Figure 6C). Mesosternum with lateral mesocoxal foveae, basisternum with longitudinal carina and transverse carina (Figure 6E). Elytra wider than pronotum (elytra:pronotum = 38.8:28.2, unit: 0.0085 mm), elytron with basal elytral sulcus and distinct subhumeral elytral fovea and sulcus (Figure 6D).

Abdominal sternite III with basolateral foveae and short median longitudinal carina, III longest, IV–VI similar length (Figure 6G). Male abdominal sternite IX exposed on VIII, female not, VIII deeply emarginated posteriorly. Abdominal tergites III–VII with basolateral foveae (Figure 6F).

Aedeagus as in Figure 9E and internal sacs of median lobe movable.

**Distribution.** South Korea (Figure 10: Diamond).

**Habitat.** Specimens of *E. lederi* were collected by sifting of soil and reed litter near horse ranches.


***Edaphus koreanus* Puthz, 2011**


(Figure 7, Figure 8F, Figure 9F and Figure 10)

*Edaphus koreanus* Puthz, 2011: 26

**Material examined.** (n = 6, 3♂♂3♀♀), 3♂♂3♀♀ (2♂♂, genitalia dissected; 1♂, slide mounted; 3♀♀, dried). Chungbuk: 1♂1♀ (1♂ genitalia dissected and mounted in Euparal on plastic card, CBNUIC), Jecheon-si, Hansu-myeon, Songgye-ri, 25 V 2021, 36°52′53.1″ N 128°05′09.6″ E, 284 m, sifting leaf litter and soil, J.-W. Kang, Y.-J. Choi, M.-H. Song; 1♀ (CBNUIC), 36°52′33.2″ N 128°05′08.6″ E, 236 m, sifting leaf litter and soil, J.-W. Kang, Y.-J. Choi, M.-H. Song; 1♀ (CBNUIC), 28 V 2020, 36°52′53.0″ N 128°05′08.0″ E, 243 m, sifting leaf litter and soil, M.-H. Song, U.-J. Byeon; Gangwon: 1♂ (1♂ genitalia dissected and mounted in Euparal on plastic card, CBNUIC), Jeongseon-gun, Bukpyeong-myeon, Sukam-ri, 21 VIII 2019, 37°29′43.0″ N 128°34′58.0″ E, 453 m, sifting leaf litter and soil near stream, M.-S. Jang, J.-Y. Kang, U.-J. Byeon; Jeonbuk: 1♂ (1♂ slide mounted, CBNUIC), Jinan-gun, 19–26 VI 2015, Lindgren funnel.

**Diagnosis.** This species can be distinguished from other *Edaphus* species by the following combination of characters: body reddish brown (Figure 7A); temples of head oblique (Figure 7B); antennomere X wider than long (width:length = 5.8:3.8, unit: 0.0085 mm), XI longer than wide (width:length = 5.9:7.1, unit: 0.0085 mm) (Figure 7K); pronotum with six basal foveae (Figure 7C), medio-basal carina about 2.3 times longer than medio-basal carina of tergite III (mbc:mbct3 = 14.1:5.9, unit: 0.0085 mm); elytron with four basal elytral foveae (Figure 7D); median lobe of male aedeagus twice longer than wide, paramere long seta 1.6 times longer than short seta (Figure 9F).

**Redescription.** Proportional measurements of habitus: HW: 27.1; DE: 18.8; LE: 7.1; LG: 21.2; LT: 24.7; PL: 30.6; PW: 32.9; dlbc: 21.2; EL: 40.0; EW: 49.4; mbc: 14.1; mbct3: 5.9 (unit: 0.0085 mm). Body length 1.1–1.3 mm. (forebody length: 0.71 mm).

Head rectangular, vertex expanded, eyes convex (Figure 7B). Antennae bearing from edge of frons, I elongate, II subrectangular, III–VI similar length, VII subqudrate, VIII–IX trapezoid, X–XI club form (Figure 7K). Maxillary palpomeres I–II elongate, III longest (Figure 7L). Labrum with seven teeth and ten setae (Figure 7M). Mandible falciform and with two setae and seven teeth at mid-level (Figure 7N).

Pronotum wider than head (head:pronotum = 27.1:32.9, unit: 0.0085 mm) and wider anteriorly and with lateral basal carina and medio-basal carina (Figure 7C). Mesosternum with lateral mesocoxal foveae, basisternum with longitudinal carina and transverse carina (Figure 7E). Elytra wider than pronotum (elytra:pronotum = 49.4:32.9, unit: 0.0085 mm), elytron with basal elytral sulcus, distinct subhumeral elytral fovea and sulcus (Figure 7D).

Abdominal sternite III with basolateral foveae and short median longitudinal carina, III longest, IV–VII similar length (Figure 7G). Male abdominal sternite IX exposed on VIII, female not, VIII deeply emarginated posteriorly. Abdominal tergites III–IX with basolateral foveae (Figure 7F).

Aedeagus as in Figure 9F and internal sacs of median lobe movable.

**Distribution.** South Korea (Figure 10: pentagon).

**Habitat.** Specimens of *E. koreanus* were collected by sifting of leaf litter and soil in the forest, Lindgren funnel also used.


**Key to Korean species of the genus *Edaphus* Motschulsky**


1 Temples somewhat bulging (Figure 2B, Figure 4B and Figure 6B)........................................................................2– Temples more or less straight, oblique (Figure 3B, Figure 5B and Figure 7B)......................................................42 HW widest; forebody length more than 0.70 mm.......................................................***......... E. haenamensis* sp. n.**– HW narrow; forebody length less than 0.70 mm.....................................................................................................33 Body color reddish brown, aedeagus (Figure 4A and Figure 9C)............................................***E. suyuensis* sp. n.**– Body color dark brown, aedeagus (Figure 6A and Figure 9E)................................***E. lederi* (Eppelsheim, 1878)**4 mbc less than twice as long as mbct3........................................................................................***E. ulsanensis* sp. n.**– mbc more than twice as long as mbct3.....................................................................................................................55 Long mbc; abdomen reddish brown; aedeagus (Figure 9D)........................................***E. koreanus* (Puthz, 2011)**– Short mbc; abdomen dark brown; aedeagus (Figure 9B)....................................................***E. odaesanensis* sp. n.**

## 4. Discussion

Species of the genus *Edaphus* are difficult to distinguish from each other. The main external diagnostic characters in this genus are the head shape, antennomeres X–XI shape, medio-basal fovea, mbc (medio-basal carina of the pronotum), and mbct3 (medio-basal carina of tergite III). The shape of abdominal sternites VIII and IX in males are used as the the main identification characters, as well as the aedeagus. The bionomics in this genus are poorly known, but most specimens were found in wet leaf litter and the upper layer of soil, sampled by sifting, and occasionally collected by soil-washing, flight intercept traps, etc. As regards the Korean fauna, because of their cryptic habits, more *Edaphus* species are expected to be found in the future.

## 5. Conclusions

This genus has approximately 600 species worldwide. Most species are distributed in the Oriental region. Previously, a single species was recorded in South Korea. In this study, four new species (*Edaphus haenamensis*
**sp.n.**, *E. odaesanensis*
**sp.n.**, *E. suyuensis*
**sp.n.**, and *E. ulsanensis*
**sp.n.**) and one unrecorded species *(E. lederi*) were recorded in Korea fauna.

## Figures and Tables

**Figure 1 insects-13-00362-f001:**
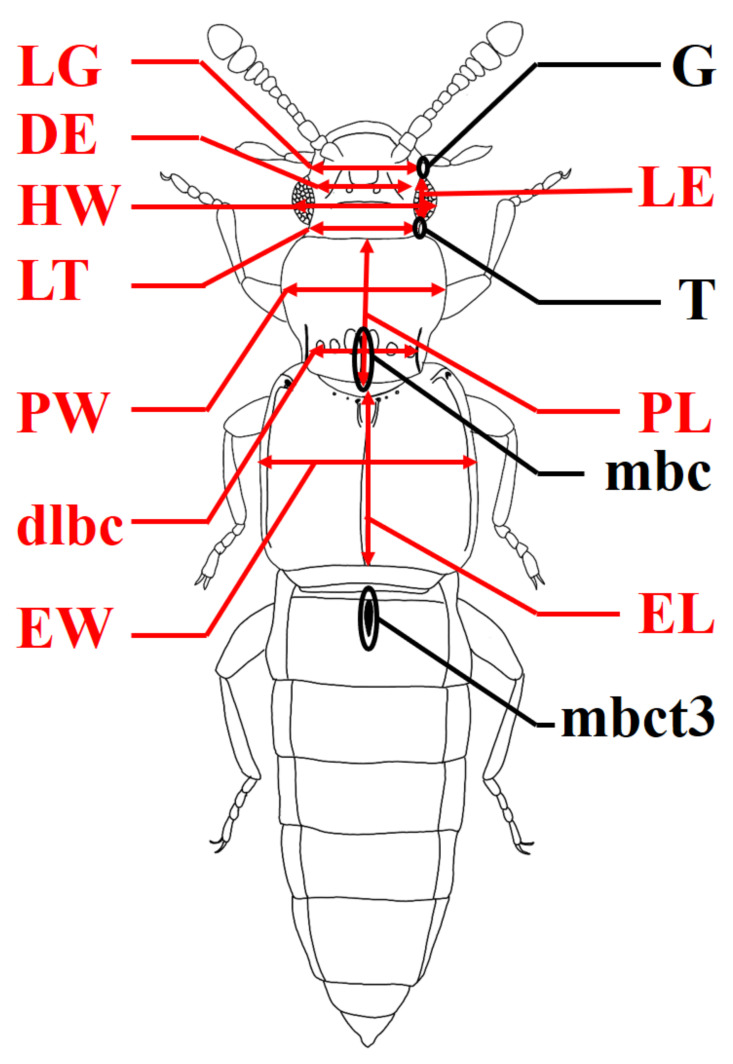
Abbreviation of morphological character: DE–distance between eyes; dlbc–distance of the latero-basal carinae of the pronotum; EL–greatest length of elytra; EW–greatest width of elytra; G–gena; HW–width of head; LE–length of eyes; LG–length of genae; LT–length of temples; mbc–medio-basal carina of the pronotum; mbct3–medio-basal carina of tergite III; PL–length of pronotum; PW–width of pronotum; T–temple.

**Figure 2 insects-13-00362-f002:**
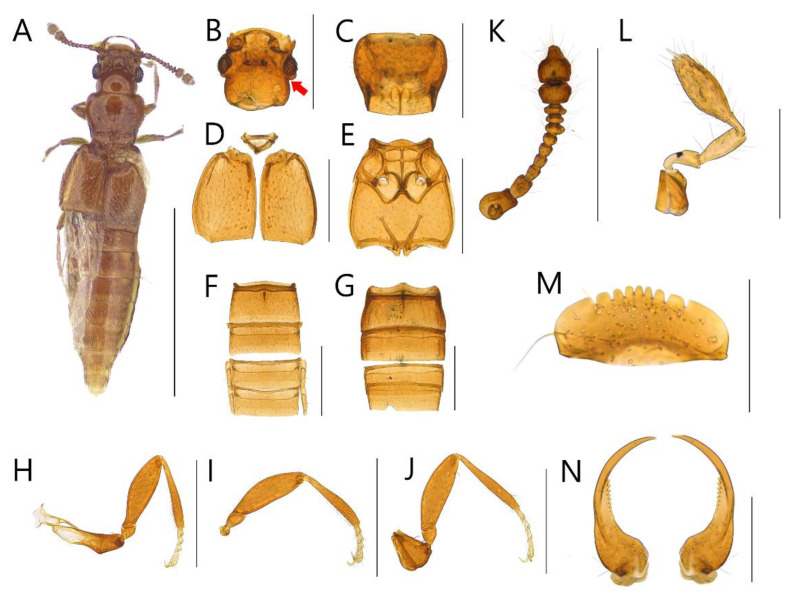
Male habitus of *Edaphus haenamensis* **sp.n.:** (**A**) dorsal view; (**B**) head; (**C**) pronotum; (**D**) elytra; I meso and metasternum; (**F**) abdominal tergites; (**G**) abdominal sternites; (**H**) fore leg; (**I**) middle leg; (**J**) hind leg; (**K**) antenna; (**L**) maxillary palp; (**M**) labrum; (**N**) mandibles. Scale bars: (**A**) = 1 mm; (**B**–**K**) = 0.3 mm; (**L**–**N**) = 0.1 mm.

**Figure 3 insects-13-00362-f003:**
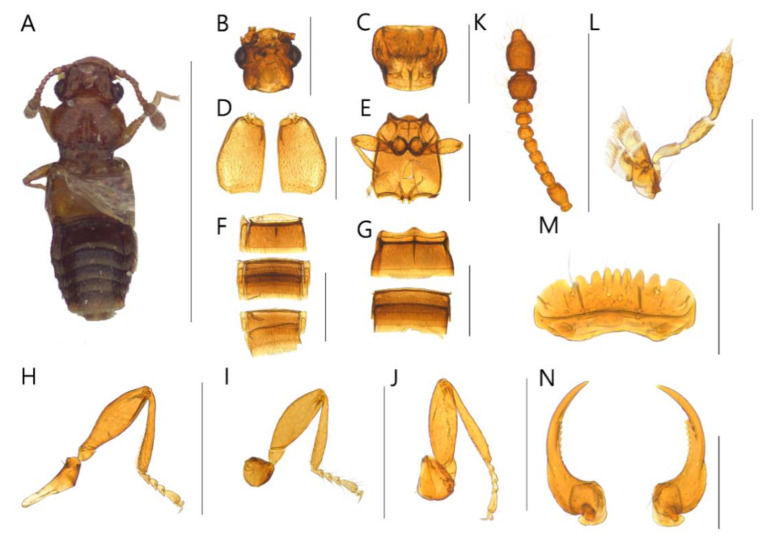
Male habitus of *Edaphus odaesanensis* **sp.n.:** (**A**) dorsal view; (**B**) head; (**C**) pronotum; (**D**) elytra; (**E**) meso and metasternum; (**F**) abdominal tergites; (**G**) abdominal sternites; (**H**) fore leg; (**I**) middle leg; (**J**) hind leg; (**K**) antenna; (**L**) maxillary palp; (**M**) labrum; (**N**) mandibles. Scale bars: (**A**) = 1 mm; (**B**–**K**) = 0.3 mm; (**L**–**N**) = 0.1 mm.

**Figure 4 insects-13-00362-f004:**
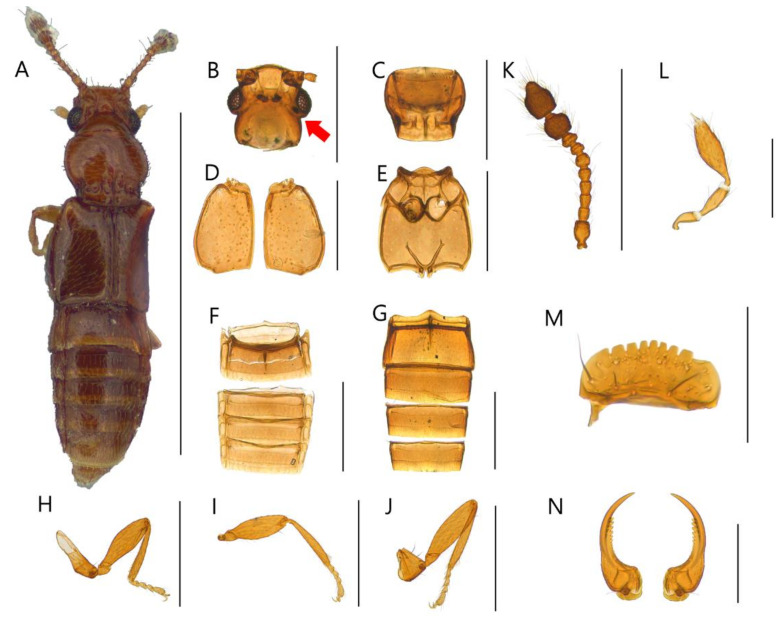
Male habitus of *Edaphus suyuensis* **sp.n.:** (**A**) dorsal view; (**B**) head; (**C**) pronotum; (**D**) elytra; (**E**) meso and metasternum; (**F**) abdominal tergites; (**G**) abdominal sternites; (**H**) fore leg; (**I**) middle leg; (**J**) hind leg; (**K**) antenna; (**L**) maxillary palp; (**M**) labrum; (**N**) mandibles. Scale bars: (**A**) = 1 mm; (**B**–**K**) = 0.3 mm; (**L**–**N**) = 0.1 mm.

**Figure 5 insects-13-00362-f005:**
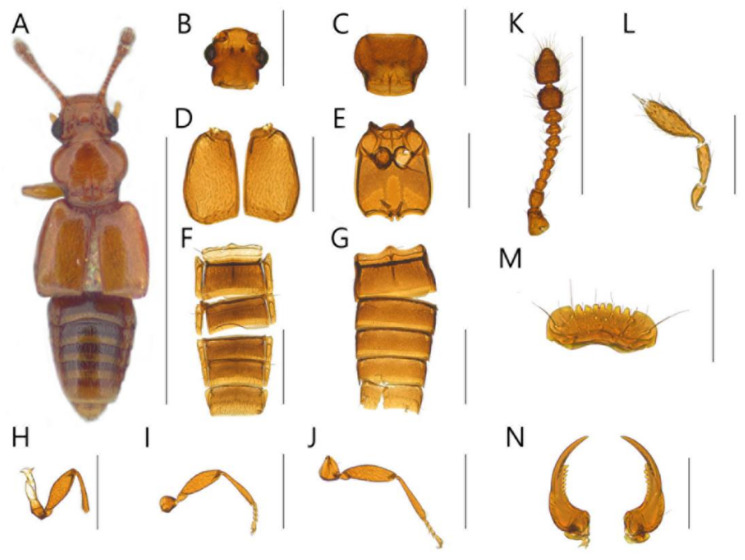
Male habitus of *Edaphus ulsanensis* **sp.n.:** (**A**) dorsal view; (**B**) head; (**C**) pronotum; (**D**) elytra; (**E**) meso and metasternum; (**F**) abdominal tergites; (**G**) abdominal sternites; (**H**) fore leg; (**I**) middle leg; (**J**) hind leg; (**K**) antennae; (**L**) maxillary palp; (**M**) labrum; (**N**) mandible. Scale bars: (**A**) = 1 mm; (**B**–**K**) = 0.3 mm; (**L**–**N**) = 0.1 mm.

**Figure 6 insects-13-00362-f006:**
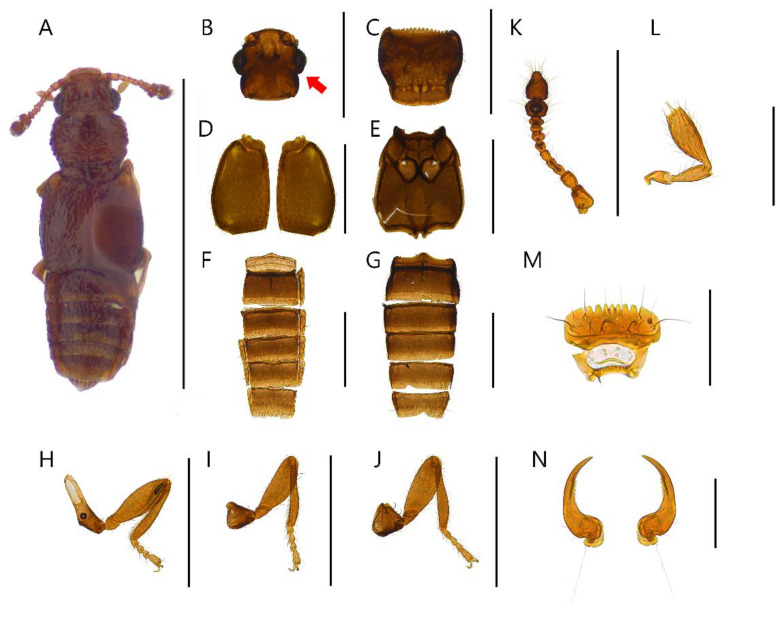
Male habitus of *Edaphus lederi* Eppelsheim: (**A**) dorsal view; (**B**) head; (**C**) pronotum; (**D**) elytra; (**E**) meso and metasternum; (**F**) abdominal tergites; (**G**) abdominal sternites; (**H**) fore leg; (**I**) middle leg; (**J**) hind leg; (**K**) antenna; (**L**) maxillary palp; (**M**) labrum; (**N**) mandibles. Scale bars: (**A**) = 1 mm; (**B**–**K**) = 0.3 mm; (**L**–**N**) = 0.1 mm.

**Figure 7 insects-13-00362-f007:**
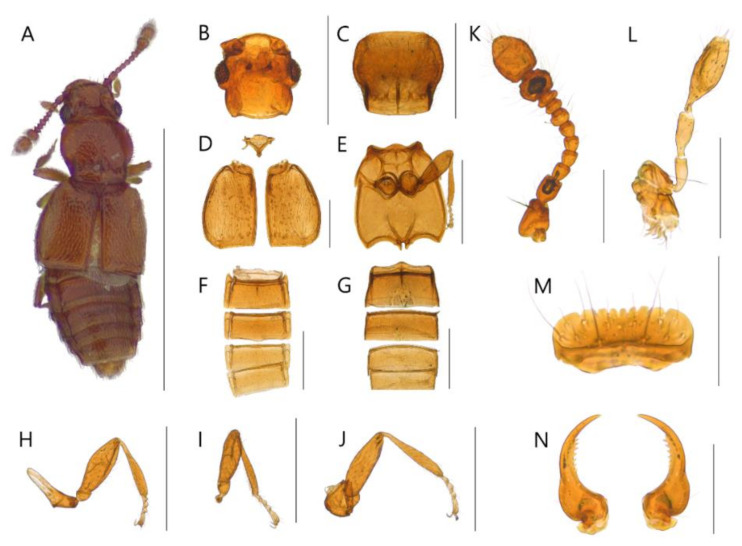
Adult of *Edaphus koreanus* Puthz: (**A**) dorsal view; (**B**) head; (**C**) pronotum; (**D**) elytra; (**E**) meso and metasternum; (**F**) abdominal tergites; (**G**) abdominal sternites; (**H**) fore leg; (**I**) middle leg; (**J**) hind leg; (**K**) antenna; (**L**) maxillary palp; (**M**) labrum; (**N**) mandibles. Scale bars: (**A**) = 1 mm; (**B**–**J**) = 0.3 mm; (**K**–**N**) = 0.1 mm.

**Figure 8 insects-13-00362-f008:**
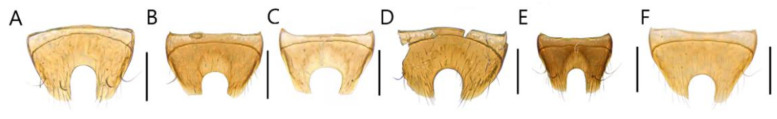
Abdominal sternite VIII of male: (**A**) *Edaphus haenamensis*
**sp.n.**; (**B**) *E. odaesanensis* **sp.n.**; **(C**) *E. suyuensis* **sp.n.**; **(D**) *E. ulsanensis*
**sp.n.**; (**E**) *E. lederi*; (**F**) *E. koreanus*. Scale bars: (**A**–**F**) = 0.1 mm.

**Figure 9 insects-13-00362-f009:**
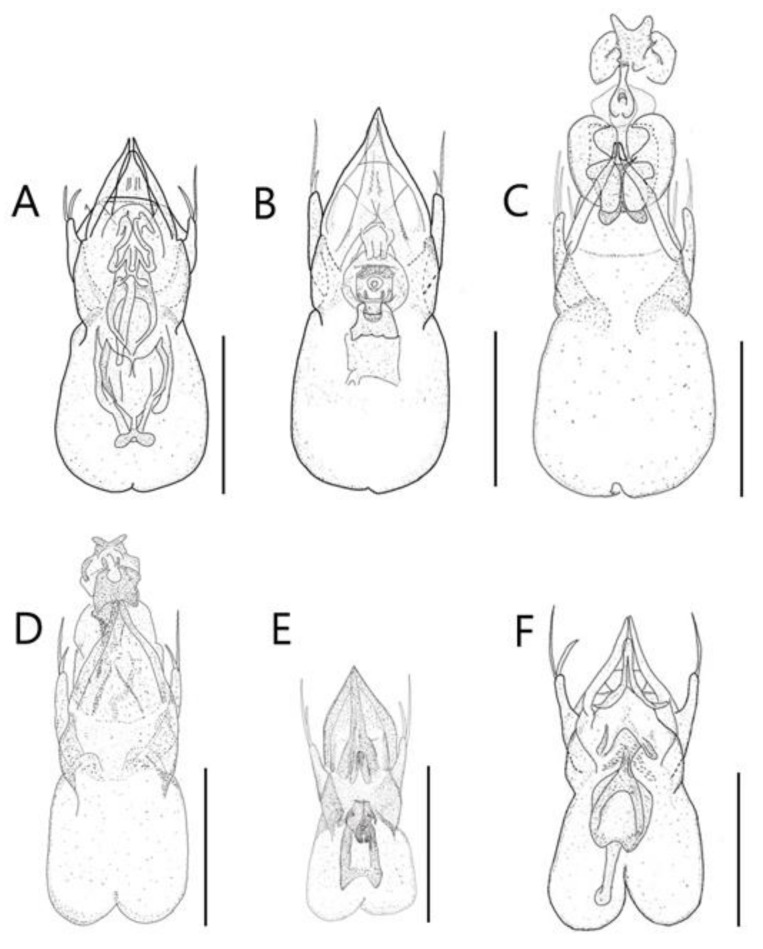
Genitalia of male: (**A**) *Edaphus haenamensis*
**sp.n.**; (**B**) *E. odaesanensis* **sp.n.**; **(C**) *E. suyuensis* **sp.n.**; **(D**) *E. ulsanensis*
**sp.n.**; (**E**) *E. lederi*; (**F**) *E. koreanus*. Scale bars: (**A**–**F**) = 0.1 mm.

**Figure 10 insects-13-00362-f010:**
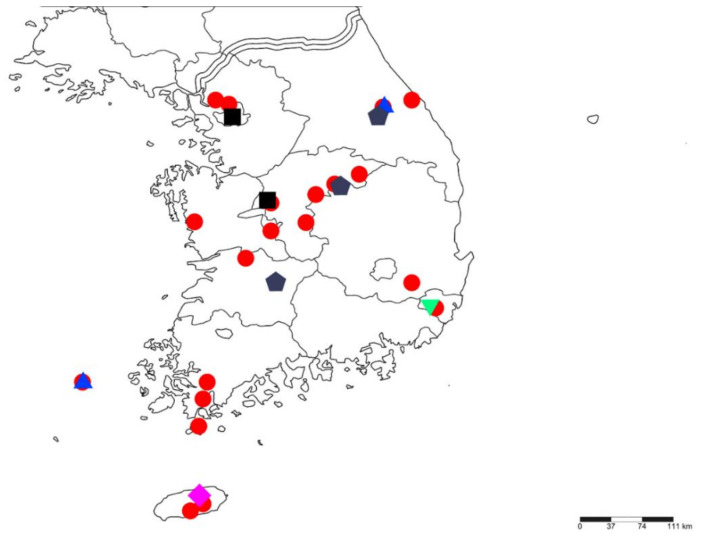
Collection localities in South Korea: *Edaphus haenamensis* **sp.n.** (**circle**); *E. odaesanensis*
**sp.n.** (**triangle**); *E. suyuensis* **sp.n.** (**square**); *E. ulsanensis* **sp.n.** (**reverse triangle**); *E. lederi* (**diamond**); *E. koreanus* (**pentagon**).

## Data Availability

This published work have been registered in ZooBank, the online registration system for the ICZN (International Code of Zoological Nomenclature).

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
