# Peer review of "Review of the Korean Species of the Genus Edaphus Motschulsky (Coleoptera, Staphylinidae) with Description of Four New Species"

_insects, 2022, doi:10.3390/insects13040362_

Round 1

Reviewer 1 Report

I would like to congratulate the authors for this nice work and would like to encourage them to continue exploring the Korean staphylinid fauna by varied methods of collecting. The goal should be achieving a state of knowledge that of Japan but by the quality of illustrations of the Shanghai Lab.

'Abstract' rewrite:

The Cosmopolitan euaesthetine genus, Edaphus Motschulsky, 1857 with about ninety Palaearctic species was formerly known by a single species, E. koreanus Puthz, 2011 of which two specimens were collected in the southern part of the Korean Peninsula. In this paper, the knowledge of the Korean Edaphus fauna is expanded to include six species, including four described here based on a rich material collected in recent years. A key to the Korean Edaphus species, illustration of the habitus and diagnostic characters, and a distribution map are provided.

'Discussion' rewrite:

Species of the genus Edaphus are difficult to distinguish from each other. The main external diagnostic characters in this genus are the head shape, antennae, mbf, mbc, and mbct3. The shapes of abdominal sternites VIII and IX in male are used as the main identification characters as well as the aedeagus. The bionomics in this genus are poorly known, but most specimens are found in wet leaf litter and the upper layer of soil, sampled by sifting, occasionally collected by soil-washing, flight intercept traps, etc. As regards the Korean fauna, because of their cryptic habits, more Edaphus species are expected to be found in the future.

I do not like the "proportional measurements". Why cannot the values be in mm? When identifying beetles measurements are often useful. It would also be good to know the variabilities of the values in mean (minimum-maximum) format and number of measured specimens given. Personally, I find the value of pronotal width the easiest to measure, it is rather consistent (but pronotal length is even more consistent), so such values (with minimum-maximum ranges) should be included in the key for every species. (The forebody length could be similarly informative but difficult to measure in poorly mounted specimens and is affected by the protrusion of the neck etc.)

I do not like the material listing either. It is not clear how many specimens were present. As a very minimum, the number of examined specimens must be given for each species. In some instances I got confused over whether the holotype was photographed then completely dissected and slide mounted (if so, this must be mentioned explicitly) and why "Paratype" is used in the singular for the list of paratypes. Inconsistencies like this must be resolved.

ILLUSTRATIONS

Fig. 1A is not suitable in this size and resolution. This figure should be somewhat larger and more clear. It is showing too many features over small space. I think many of these features are obvious to the reader, so basically it is not presenting anything important except for the abbreviations of features which may be done in the main text.

Figs 1B and C are not very informative as the sternites do not have any special modification except for sternite VIII in the male. I think these figures should be deleted.

The 'habitus figures' (Fig. 2A, 3A, etc.) - if one does not make the slightest effort to present these specimens in a more eye-pleasing way, these figures should be deleted. I cannot believe that specimens mounted so badly are shown (lighting of these photos is really poor!) and in the age of Photoshop better effort cannot be made to make these images more consistent and comparable. In the present state they are a distraction and a shame. In general, a feature to be shown must be set so that the viewpoint is perpendicular to its main axis/plane. If this cannot be achieved, the photo should not be taken. If the purpose of these images is to document the (depressing) condition of the specimens, they should be put on a website.

It is unfortunate the the internal sacs of E. suyuensis and E. ulsanensis aedeagi are everted. With a single specimen available this cannot be cured, but if more are present, I'd attempt a second drawing. According to my taste, adeagus drawings should be larger.

Reviewer 2 Report

The manuscript titled “Review of the Korean species of the genus Edaphus Thomson (Coleoptera, Staphylinidae) with description of four new species” refers to Edaphus, cosmopolitan genus of Euaesthetinae, which contains extremely small beetles, that makes their study rather difficult. The present research is a part of the world history of Edpahus fauna, over 250 species have been described in the last 20 years. The Edpahus fauna of the Korean Peninsula is extremely poorly known, thus, new information presented in this work is very needed and valuable. The manuscript contains all element typical for studies in the field of ​​alpha taxonomy. You can see that the Authors have experience in presenting that type of results. Descriptions of morphological structures of new species are generally exhaustive and clearly presented. The color photos, corresponding perfectly with the descriptions, are valuable aspect of the paper. Therefore, the manuscript should be published in the Insects journal but after considering, the following suggestions:

  1. Title: Line 2: „ … Edaphus.” should be “…Edaphus Motschulsky …”
  2. Materials and Methods: line 56-59: the following acronyms were used: “HW–width of head, LE–length of eyes, LG–length of genae, LT–length of temples” consistently should be: EL–length of eyes; GL–length of genae TL–length of temples
  3. Results:

Generally: In the characteristics of each new species, further details of their habitats, if possible, could be given (e.g.forest or open area etc.)

Line 76-79: The Authors list diagnostic characters of the genus. This suggests that the data is new, selected by the Authors. However, in some articles, for example: Kistener 1962: A Revision of the Nearctic and Ethiopian Species of the Genus Edaphus (Coleoptera: Staphylinidae), we find a set of generic features, including the 4-4-4 tarsal formula, and others, which are also provided by the Authors. I propose, to take into account the literature data and quote them.

Line 159-160, 204, 249, 287, 323, 368: the abbreviations “EL …EW” that symbols are not explained in Materials and Methods

Line 288: „ Body length 1.1–1.4 mm”. A size range appear but the species Edaphus ulsanensis is represented by only single specimen.

Line 314, 359: Since the species Edaphus lederi and E. koreanus are known to science “description” should be replace by redescription.

Key to Korean species of the genus Edaphus

The key is a little controversial. It is uncertain whether color features are strong data in species diagnostics. Especially, they are the only ones, distinguishing E. lederi from the other 5 species. Also, the body size metric features of E. haenamensis (1.2 ~ 1.7 mm) and E. suyuensis (1.0 ~ 1.3 mm) are not clear. Helpful hints on what features can be useful are given in Putz (2014) in the introduction, as well as in Putz (2010) in the key for the determination of 21 species from Japan.

Figure 1 A

This figure is helpful and necessary, but not very readable (overlap lines, small symbols).

Discussion

That part is not a real discussion. The Authors could refer to literature data on two known species E. lederi and E. koreanus (Putz 2014, Webster et al. 2016 and others).

Reviewer 3 Report

I would reccomend to add informations in the introduction, according to comments in pdf. Please, rearrange contents of chapter Discussion and Conclusion, with developing statements in Conclusion - all comments if pdf file.

Reviewer 4 Report

This is a good solid taxonomic manuscript using modern methods and normal species level recognition characters. Confidence in the species limits is improved because the opinion of a world authority was obtained. The characters are subtle, and the key could be improved to include better description comparisons. The aedeagal characters in the key need to be described in addition to referencing the illustrations. Sometimes relevant characters are not obvious in the illustrations. The English text needs to be proofed and some minor corrections are required for better English presention. Suggest getting a native English writer to go over it. 
